# Growth Substrate Geometry Optimization for the Productive Mechanical Dry Transfer of Carbon Nanotubes

Andre Butzerin [1,*] , Sascha Weikert [2] and Konrad Wegener [2]

1 Institute of Machine Tools and Manufacturing, ETH Zürich, 8092 Zürich, Switzerland
2 inspire AG, 8005 Zürich, Switzerland
* Correspondence: butzerin@iwf.mavt.ethz.ch

**Abstract:** The selection of growth substrate geometries for the mechanical dry transfer of carbon nanotubes to device substrates depends on the precision of the assembly equipment. Since these geometries play a decisive role in the overall efficiency of the process, an investigation of the most important geometry parameters is carried out. The substrate geometry affects the number of carbon nanotubes suspended during the growth process and the speed of mechanical assembly at the same time. Since those two criteria are interlinked and affect productivity, a meta-model for the growth and selection of the nanotubes is simulated and a time study of the resulting assembly motions is subsequently performed. The geometry parameters are then evaluated based on the total number of suspended carbon nanotubes and the throughput rate, measured in transfers per hour. The accuracy specifications are then taken into account. Depending on the overall accuracy that can be achieved, different offset angles and overlaps between the growth and receiving substrate can be reached, which affect productivity differently for different substrate geometries. To increase the overall productivity, growth substrate designs are adapted to allow fully automated operation. This measure also reduces the frequency of substrate exchanges once all carbon nanotubes have been harvested. The introduction of substrates with multiple, polygonally arranged edges increases the total number of nanotubes that can be harvested. The inclusion of polygonally arranged edges in the initial analysis shows a significant increase in overall productivity.

**Keywords:** optimization; productivity; mechanical dry transfer; substrate geometry; suspended carbon nanotube

## 1. Introduction

The mechanical dry transfer method of placing carbon nanotubes onto substrates is known to be very clean and therefore results in exceptional device performance. With this type of production process, a distinction can be made between deterministic and non-deterministic placement. With the non-deterministic placement, as reported in [1], carbon nanotubes are scattered randomly over a device substrate. With this method, a very high number of transfers per hour (TPH) can be achieved, but there is no control over the position and orientation of the individual carbon nanotubes. The deterministic mechanical dry transfer manufacturing technique of carbon nanotube devices, on the other hand, utilizes a growth substrate as a carbon nanotube donor to integrate those into a device substrate. The growth substrate has cantilever pairs that form trenches in which a carbon nanotube can be suspended. In [2–9], nanotubes are suspended on such cantilever pairs and are transferred to the device substrate by mechanically breaking them off their support structure. Van der Waals forces guarantee adhesion to the device electrodes. This procedure is schematically depicted in Figure 1. As, however, each single carbon nanotube has to be transferred individually, the achievable production rate is a severe disadvantage. Moreover, as it is necessary to use a motion system for the positioning of carbon nanotubes, travel distances and ranges need to be considered as well.

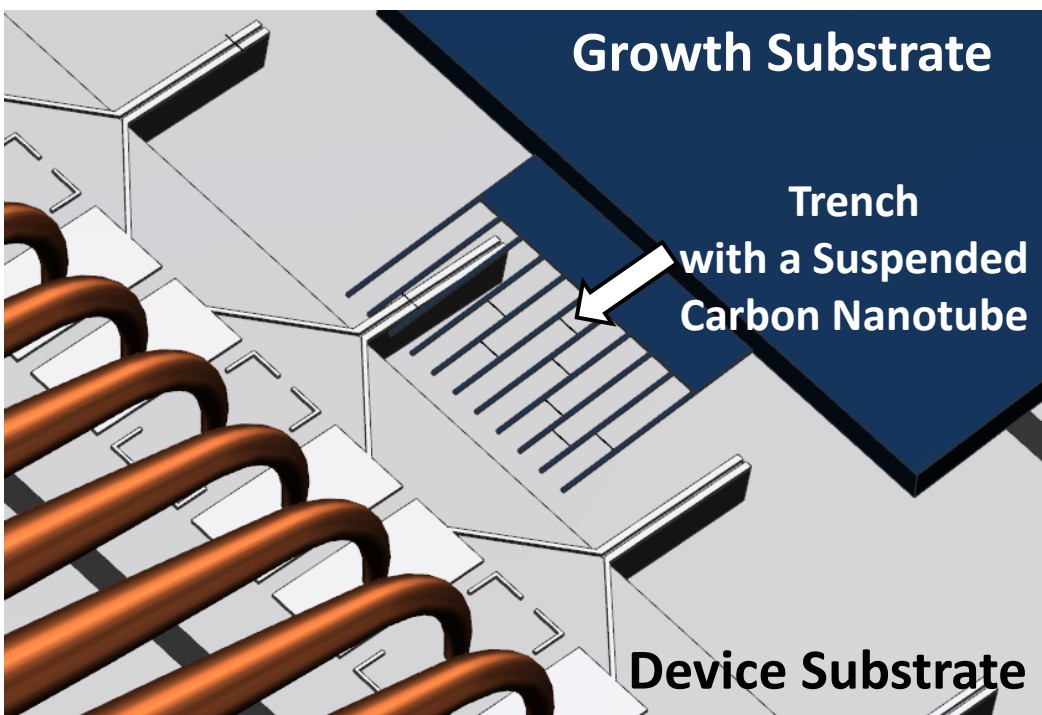

**Figure 1.** Visualization of growth and device substrate interaction during mechanical dry assembly. Carbon nanotubes are transferred by breaking them off from the cantilevers. They are held in place on the device substrate by Van der Waals forces. After each transfer, the next carbon nanotube on the growth substrate is moved to the next electrodes on the device substrate and the process is repeated.

The growth substrate's geometry also affects the carbon nanotube synthesis. During the synthesis procedure, the number of nanotubes which are suspended across the cantilevers is determined. In addition, their relative angle to the substrate and their actual length are defined as well. Depending on the application, it can be desired to have multiple carbon nanotubes on a single device. As the number of tubes which are suspended on the same pair of cantilevers varies, a selection regarding the desired number of tubes per trench (TPT) is required too. Hence, substrate geometries must be optimized to increase the production rate while taking effectiveness of growth, selection and assembly into account.

Growth substrates are already commercially available and do not have to be manufactured in-house. However, their use does not address large batch fabrication. As shown in [8], only eight devices are fabricated with a growth substrate of 48 cantilevers and a cantilever pitch of 60 μm.

The selection of the optimal type of growth substrate geometry can be based on a multi-criteria decision-making model [10,11]. The approach presented in this work, however, focuses on how the growth substrate geometry parameters affect both productivity and accuracy throughout the whole process chain, from synthesis to assembly, using simulation.

## 2. Productivity

The geometry of a substrate has a significant impact on the overall process efficiency. In addition to the growth parameters, the geometry of the part where the nanotubes are suspended on the substrate dictates the density of harvestable tubes. Hence, the achievable productivity is mainly characterized by these features besides the growth process requirements. Therefore, the process scheme shown in Figure 2 is carried out. A simple simulation of nanotube growth is introduced which considers the cantilever geometries and the growth density of the CVD process. The output of this simulation yields a list of the coordinates of each transferable nanotube. Experimentally grown carbon nanotube substrates validate the proportion of harvestable tubes determined by simulations and the parameters used. The resulting tube list serves, together with the substrate geometry and

trajectory parameters of the manipulator, as input for a consecutive time study. The final result is a transfer rate in transfers per hour (TPH) for a given substrate geometry that is achievable with a manipulator's positioning performance. Furthermore, hints about the replacement rate of the growth substrate are given by the number of transferable trenches.

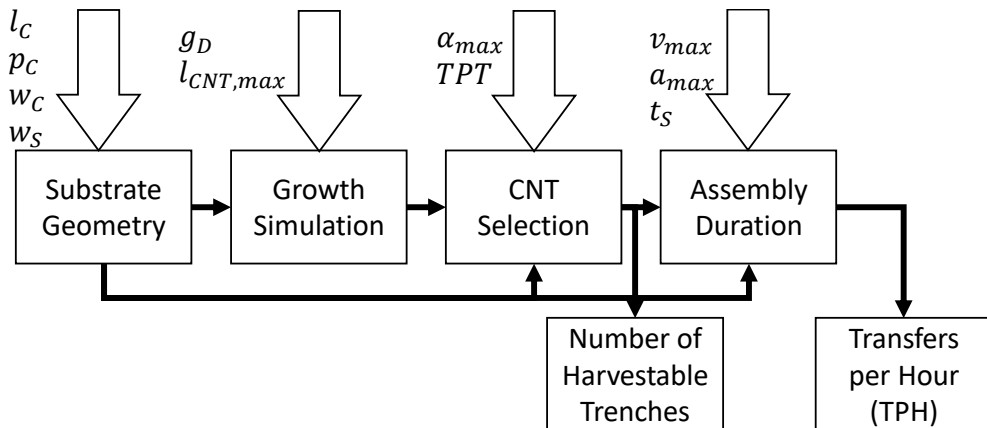

**Figure 2.** Analysis scheme of growth substrate geometry parameters for the simulation of productivity.

### 2.1. Simulation of Nanotube Growth and Selection

The first step that is required for the growth simulation is the substrate geometry. The simulation only considers a two-dimensional substrate geometry in order to keep the level of complexity low. The input parameters for the creation of the substrate geometry are the length $l_C$ and width $w_C$ of the cantilevers and their pitch $p_C$ as shown in Figure 3. With these parameters given and additional knowledge of the total substrate edge width $w_e$, sufficient information is given to create the desired substrate. As shown in Figure 3, the resulting number of cantilevers $n_C$ for carbon nanotube growth can be calculated as

$$n_C = \frac{w_e - w_C}{p_C} + 1 \tag{1}$$

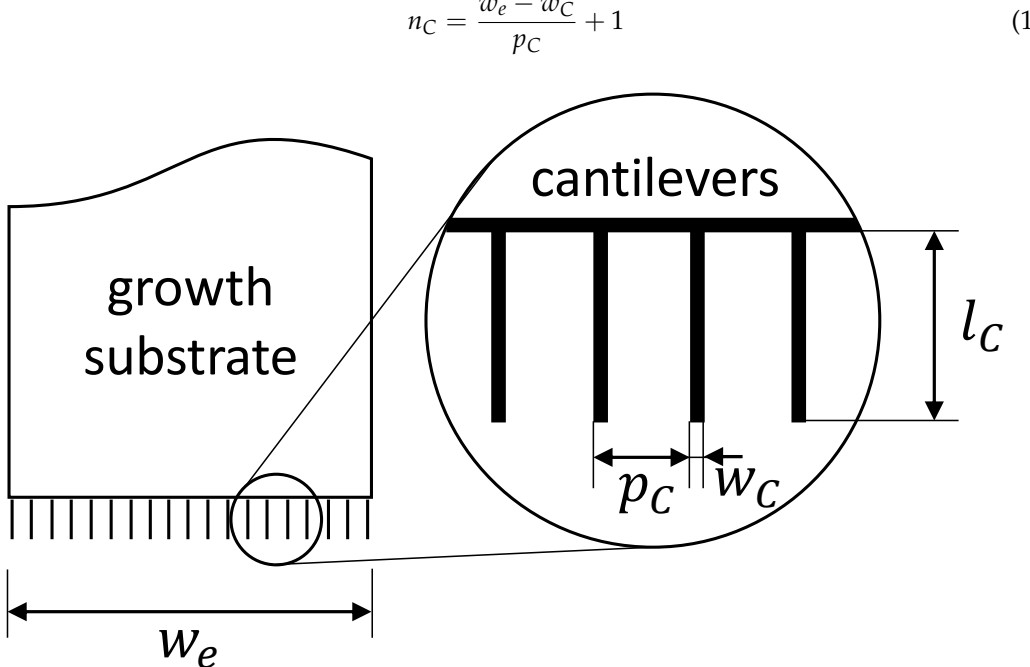

**Figure 3.** The crucial cantilever geometry of a comb-like growth substrate used for mechanical dry transfer.

After the determination of the substrate geometry the growth can be simulated. Additional required parameters for this are the maximum length $l_{CNT,max}$ of carbon nanotubes

to be grown and the growth density $g_D$ with which they are distributed across the substrate. The growth density $g_D$ therefore denotes the density of the carbon nanotubes of the desired type, such as semiconducting and/or metallic.

For the sake of simplicity, it is assumed that grown carbon nanotubes are always straight and that they are defect-free. Only their length, position and orientation is required to determine the location of the cantilever pair between which they are suspended. In order to reduce computational load, the growth area of simulated nanotube synthesis is constrained to twice the cantilever length $l_C$ times the substrate edge width $w_e$ as depicted in Figure 4 in light green.

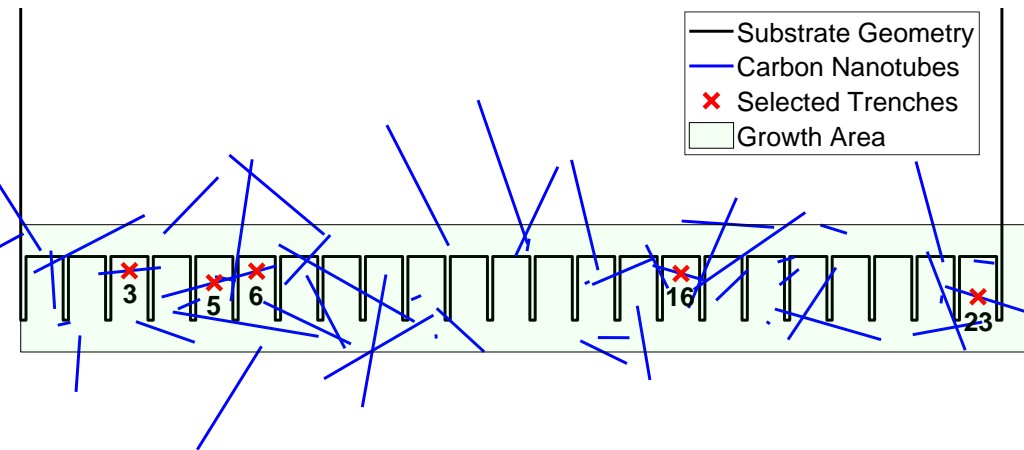

**Figure 4.** Example visualization of random carbon nanotube growth with a nominal growth density of $g_D = 3000 \frac{tubes}{mm^2}$ on one single substrate edge. The chosen substrate geometry is $n_C = 24$, $p_C = 14$ µm, $w_C = 2$ µm, $l_C = 24$ µm. The tube angle tolerance $\alpha_{max}$ for trench selection is $\pm 20°$ and the desired number of suspended tubes per trench $TPT = 1$.

The simulation considers small variations in growth density via uniform distribution and over a sample size of $n_S$ in total. Those affect the position and orientation of the suspended nanotubes. Larger variations in growth density resulting from experimental imperfections, however, have not been considered.

The tubes designated for transfer can be determined once there is a given substrate geometry with carbon nanotubes distributed across its cantilevers. At this step a distinction between tubes that are suspended and tubes that are not suspended is made. A tube is considered suspended if it has at least two intersections with the contour of the substrate geometry that are at least one nominal trench width ($p_C - w_C$) apart along the tube axis. All other tubes are disregarded.

Since multiple carbon nanotubes can be suspended on a single pair of cantilevers, it is important to categorize each trench by the count of tubes per trench $TPT$. This is required in order to be able to filter the trenches later by this number, as it may be necessary to assemble devices with a specific number of carbon nanotubes.

In Figure 4, the trenches chosen for transfer are indicated with a red "x". It is the outcome of a filter criterion of one carbon nanotube per trench $TPT = 1$ with a maximum relative angle $\alpha_{max} = \pm 20°$. The resulting list of tubes contains an address table of all trenches to be transferred.

### 2.2. Time Study

With the tube list as information it is feasible to derive an expected productivity of the assembly. The result is quantified in transfers per hour ($TPH$) and is a measure of the achievable productivity with the parameters applied. It can be calculated based on the substrate geometry and trajectory parameters. Assuming an acceleration limited trajectory

over a distance $D$ and a maximum acceleration $a_{max}$, the duration $\Delta t$ for a positioning step is calculated depending on whether maximum velocity $v_{max}$ can be reached or not. If

$$D \geq \frac{v_{max}^2}{a_{max}} \tag{2}$$

is true, maximum velocity $v_{max}$ is reached. For this case the duration is calculated as

$$\Delta t = 2 \cdot \frac{v_{max}}{a_{max}} + \frac{D - \frac{v_{max}^2}{a_{max}}}{v_{max}} \tag{3}$$

and for the case where maximum velocity $v_{max}$ is not reached the duration is calculated as

$$\Delta t = 2 \cdot \sqrt{\frac{D}{a_{max}}} \tag{4}$$

The sequence of $X$, $Y$ and $Z$ motions for one assembly cycle are taken from [12] where the required travel distances are calculated according to the addresses in the tube list and the substrate geometry. In order to have sufficient statistical significance, the simulation is repeated $n_S$ times for one type of substrate geometry. Each type of substrate geometry has $n_S$ of differently distributed sets of carbon nanotubes that are suspended between the substrate's cantilevers. From these simulation samples, an average transfer rate is calculated from the ratio of the total number of assembly cycles that have been performed and the sum of their duration. The number of trenches with the specified amount of tubes per trench $TPT$ is tracked as well. This value can be associated with the frequency of growth substrate replacements.

In Figure 5, the productivity $P$ in terms of $TPH$ and number of harvestable trenches $n_{harv}$ with $TPT = 1$ is shown over the geometry parameters of the cantilever pitch $p_C$ and the cantilever length $l_C$. While the productivity in form of $TPH$ only takes manipulator movements into account, the replacement rate of the substrate itself is considered by the number of harvestable trenches $n_{harv}$. The substrates need to be replaced less frequently if the number of harvestable trenches $n_{harv}$ is high.

Conclusions from Figure 5a,c show that substrate geometries with narrow cantilevers are preferable. The overall productivity is higher since substrates with such a cantilever geometry offer a higher trench density. This leads to a higher number of harvestable trenches $n_{harv}$ while at the same time the travel distance is shortened. In contrast, and as it can be seen in Figure 5b,d, the cantilever length $l_C$ is limited to the lower and upper end. Short cantilevers cannot suspend as many carbon nanotubes and long cantilevers suspend too many carbon nanotubes. Hence, the optimum cantilever length lies somewhere in between and depends on $TPT$ and the growth parameters chosen.

The results from Section 2 stipulate to minimize the cantilever pitch $p_C$ for a maximum productivity, while adjusting the cantilever length $l_C$ to a desired carbon nanotube growth. However, due to the finite accuracy of the manipulator, which has to move the growth substrate for the nanotube transfer, the cantilever pitch $p_C$ is limited on the lower end.

The simulation results of the cantilever geometry used in [8] show a median productivity of 128 $TPH$ for the same simulation parameters only with an adapted maximum nanotube length $l_{CNT,max}$ = 120 μm. This clearly shows the potential for optimizing growth substrate geometries for large-scale production.

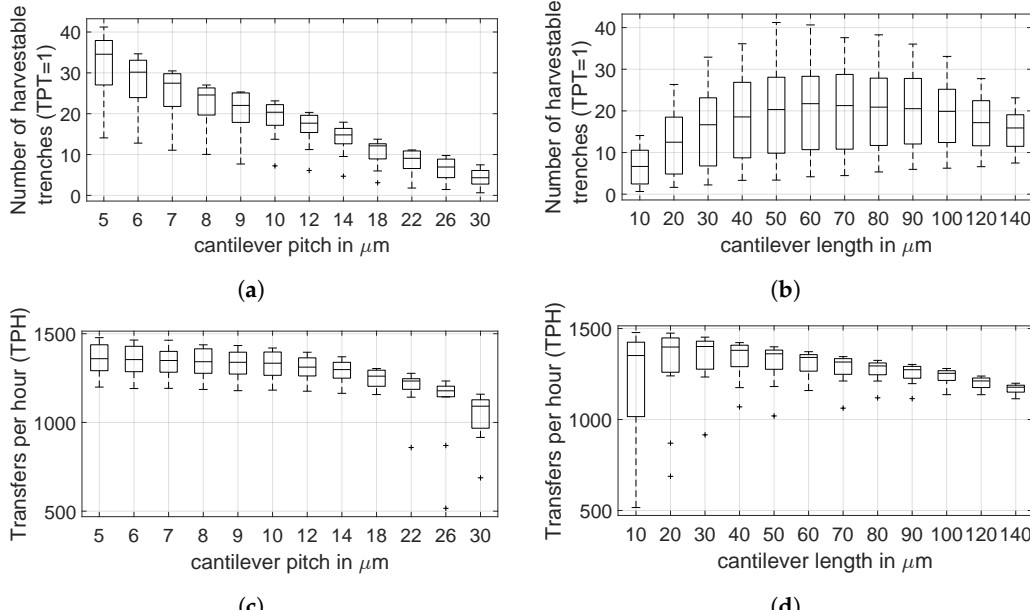

**Figure 5.** Simulation results for 144 substrate geometry variations with the according number of harvestable trenches $n_{harv}$ where $TPT = 1$ and the $TPH$ over the cantilever length $l_C$ and cantilever pitch $p_C$. Each $p_C$ is evaluated for all $l_C$ and vice versa. Boxes represent 25th and 75th percentiles, whiskers extreme values and outliers are plotted individually. Each geometry has a cantilever width $w_C = 2$ μm and was simulated $n_S = 50$ times. Used parameters for growth are $g_D = 10,000 \frac{tubes}{mm^2}$ and $l_{CNT,max} = 40$ μm. The angle tolerance $\alpha_{max}$ for tube selection was set to $\pm 10°$. Results presented are the outcome of a polygonal substrate with a number of edges $n_{edges} = 2$, the effects of which are discussed in Section 3. (**a**) As one nanotube can be suspended across multiple trenches, a reduction in cantilever pitch $p_C$ increases the number of harvestable trenches $n_{harv}$. (**b**) The optimal cantilever length $l_C$, here 60 μm, is impacted by the maximum nanotube length $l_{CNT,max}$ used in the growth simulation. (**c**) With decreasing cantilever pitch $p_C$, the number of $TPH$ increases as the travel distance is shortened and the trench density becomes higher. (**d**) The longer the cantilevers are, the higher the required travel distance becomes, causing a lower $TPH$.

### 2.3. Clearance Due to Geometry and Accuracy

Figure 6 depicts the schematic trench and device geometry during transfer. The clearance $c$ is the horizontal minimal distance between both substrates, $w_D$ is the device width and the errors $\Delta x$ and $\Delta y$ are the total positioning errors along the x and y coordinate axes. Those include errors from the manipulator and measurements errors, but also the error due to imprecise substrate fabrication. Depending on the application it may be desirable to manufacture devices with aligned carbon nanotubes. If this is the case the inclination angle $\Theta$ has to correct for the angle of the tube. The error $\Delta\varepsilon$ of $\Theta$ must then also be taken into account. Based on these considerations, the available clearance $c$ can be defined as

$$c = \frac{p_C - w_C - w_D}{2} - (y_O + \Delta y) \cdot |\tan(\Theta + \Delta\varepsilon)| - |\Delta x| \tag{5}$$

If the positioning and preparation of the growth substrate is inaccurate, the transfer of carbon nanotubes becomes less efficient and less reliable, as the probability of collisions between the two substrates is higher. With a comparison of coefficients in Equation (5), it can be seen that $\Delta x$ impacts clearance the most. The angular deviation $\Delta\varepsilon$ and the deviation in $\Delta y$ are rather uncritical. However, if those deviations are underestimated, it becomes apparent that the two substrates are likely to crash with another. Depending on the magnitude of error, it can happen such that only individual cantilevers break off or even the whole substrate.

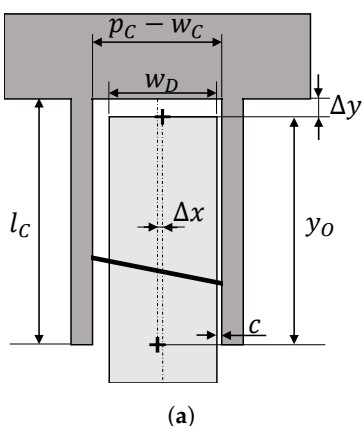 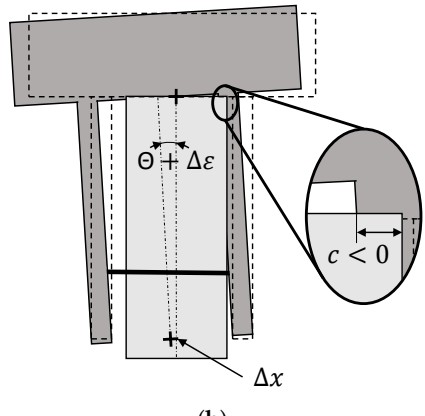

(**a**)                                       (**b**)

**Figure 6.** Geometries for the growth substrate in dark gray and the device in light gray. A carbon nanotube, depicted between growth substrate cantilevers in black is transferred to the device by moving perpendicular to the image plane. The offset is shown by the magnitudes of $\Delta x$, $\Delta y$ and $\Delta \varepsilon$. (**a**) The existing clearance $c$ depending on substrate geometries, overlap $y_O$ and position errors $\Delta x$ and $\Delta y$. (**b**) Negative example with collision of substrates. The alignment error $\Delta \varepsilon$ and the overlap $y_O$ are too large to avoid collision, respectively.

As collision must be avoided at all times, $c$ must in any case be greater than zero. It is not required to have an overlap $y_O$ equal to the full length $l_C$ for a successful transfer. Hence, the clearance along the y direction is chosen such that it is equal to the maximum expected error $\Delta y$. However, the number of harvestable trenches $n_{harv}$ decreases proportionally with $y_O$ as carbon nanotubes located deeper in the trench cannot be reached. The consequence is a productivity reduction (TPH) by a factor of $\eta_O$.

$$\eta_O = \frac{y_O}{l_C} \tag{6}$$

With a more precise manipulator and accurate substrate fabrication, $y_O$ can be increased because of a smaller error margin $\Delta y$.

For the case of automatized carbon nanotube assembly, where device after device is approached like in Figure 6, the overshoots have to be taken into account. In [12], the average overshoots $\bar{x}_{os}$ and $\bar{y}_{os}$ for a parallel kinematic micromanipulator are 0.3 µm for the x axis and 0.8 µm for the y axis, respectively. Assuming perfect substrate fabrication and negligibly small measurements errors, it can be concluded that $\Delta x \approx \bar{x}_{os}$ and $\Delta y \approx \bar{y}_{os}$. Following this assumption, a family of curves, as in Figure 7, shows the transfer parameters overlap $y_O$ over $\Theta$ where the clearance $c = 0$.

### 2.4. Implications to Productivity

Analogous to Figure 7, the accuracy and the substrate geometry affect the maximum possible overlap $y_O$. The maximum possible overlap for a given accuracy and geometry can be calculated by inserting $c = 0$ in

$$y_O = \frac{\frac{p_C - w_C - w_D}{2} - |\Delta x| - \Delta y \cdot |\tan(\Theta + \Delta \varepsilon)| - c}{\tan(\Theta + \Delta \varepsilon)} \tag{7}$$

As it can be seen from Equation (6), $\eta_O$ is also a function of $y_O$ and the cantilever length $l_C$. Since the different geometry affects productivity (TPH) and the overlap factor $\eta_O$ alike, the cantilever geometries need to by chosen according to $\Theta$ in Figure 8 for the desired overlap to reach maximum productivity.

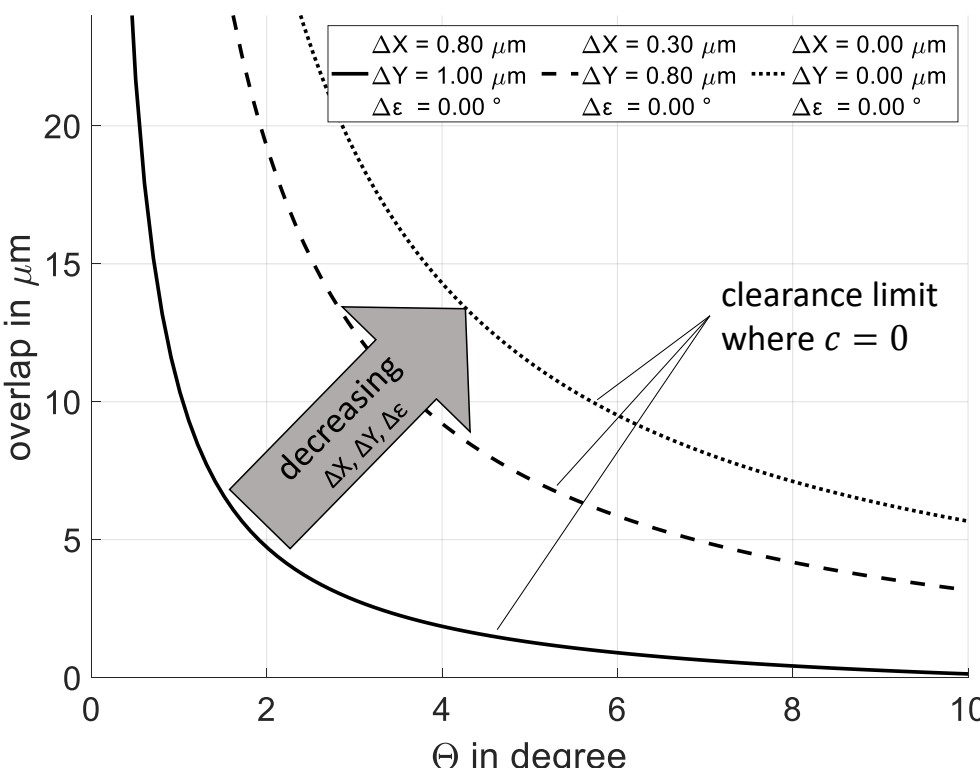

**Figure 7.** Three examples according to Figure 6 with different errors $\Delta x$ and $\Delta y$ where clearance $c = 0$. Parameter sets of overlap and $\Theta$ below each curve maintain $c > 0$. The alignment error $\Delta \varepsilon$ shifts graphs to the left along the x axis. The dotted curve highlights the maximum possible overlap $y_O$ with cantilever geometry $p_C = 14$ μm, $w_C = 2$ μm, $l_C = 24$ μm and device width $w_D = 10$ μm. The solid and dashed curves show the maximum possible overlap $y_o$ with the same substrate geometry and error values reported in [12].

Considering the geometric accuracy constraints from Equation (5), it is apparent that for $|\Theta| \approx 0$, the total error $|\Delta x|$ shows the highest sensitivity for a given substrate geometry. Therefore, $|\Delta x|$, which is the sum of the individual errors of overshoot $x_o$, static positioning of the axis $x_p$, substrate fabrication $x_f$ and re-referencing $x_{ref}$, must be kept low by machine design or tight fabrication tolerances. Otherwise, re-referencing of relative substrate positions or mapping is required. With re-referencing, the relative distance between the two substrates can be measured very close to the tool center point by optical means and thus allowing an appropriate compensation. However, this approach can also lead to an error $x_{ref}$, which must be taken into account.

As the accuracy demands to the x axis is more crucial, the overshoot error $x_o$ should be targeted to be minimized first. This means damping plays an important role, but also the moved masses, which affect the eigenfrequencies, take substantial influence. To reduce the moved mass of the x axis, it is placed on top of the axes stack.

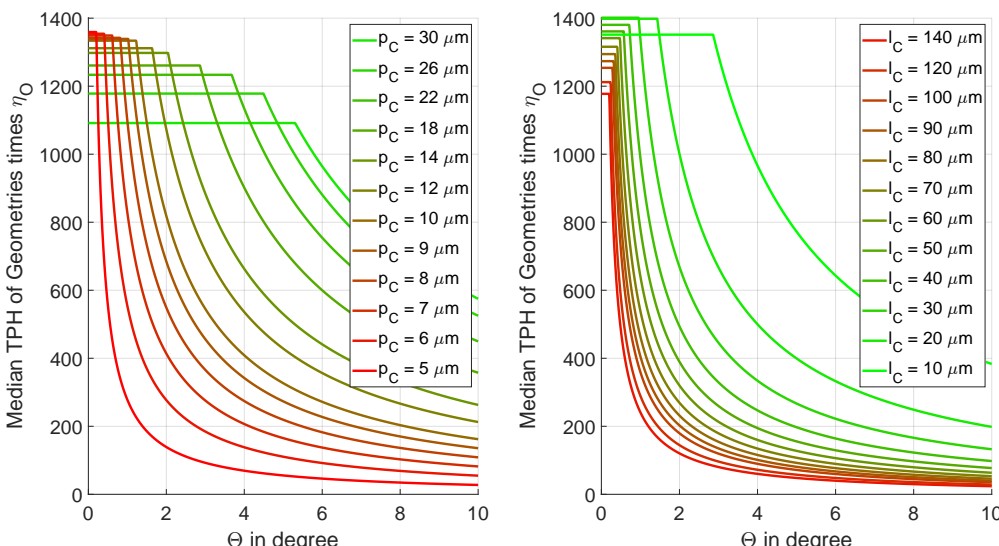

**Figure 8.** Effective productivity (TPH) of each substrate geometry parameter from the median transfers per hour of Section 2 with overlap factor $\eta_o$ taken into account. Geometry parameters $w_C$ and $w_D$ are constant for both plots. Left plot shows the effect of cantilever pitch $p_C$ on the effective productivity for a constant cantilever length $l_C = 140$ μm. The right plot shows the effect of cantilever length $l_C$ on the effective productivity for a constant cantilever pitch $p_C = 5$ μm. Both evaluations assume that errors $\Delta x = \Delta y = 0$ μm and $\Delta \varepsilon = 0°$.

## 3. Design for Automation

With the introduction of fully automated assembly, it is either required to choose a substrate geometry which offers sufficient tolerance or add a re-referencing mechanism to compensate for position errors. Beyond that, the number of harvestable trenches $n_{harv}$ is the decisive parameter for the frequency of substrate exchanges. The higher their number, the less frequent substrate exchanges are required. On top of cantilever geometry optimization, substrates with multiple edges can be introduced to decrease the required time until an exchange. This enhances the time a substrate can be used but also requires an additional rotation axis. Therefore, the total duration of the positioning motion has to be considered for an additional axis with its jerk, acceleration, velocity, travel distance and settling time. Based on this consideration, the number of substrate edges $n_e$ has to be optimized for a corresponding rotation axis. Since the substrates are of polygonal shape, the circumcircle diameter $d$ can be chosen as their size factor. Increasing the number of harvestable trenches $n_{harv}$ by increasing the diameter $d$ of the growth substrate would reduce the total number of substrates that can be produced in one batch. Furthermore, as the diameter of the growth substrate increases, the distance between the axis of rotation and the tool center point increases, resulting in a longer settling time and worse accuracy.

The number of substrate edges $n_e$ influences the edge width $w_e$ and the angular travel distance to bring the next edge into position.

$$w_e = d \cdot \sin \frac{\pi}{n_e} \tag{8}$$

The value of $w_e$ can also be calculated after Equation (1) with the number of cantilevers per edge $n_C$ and their respective geometry parameters $p_C$ and $w_C$.

$$w_e = (n_C - 1) \cdot p_C + w_C \tag{9}$$

As shown in Figure 9, the diameter $d$ of the circumcircle is selected as a constant in order to make substrates of different shapes comparable to another. Continuing from this assumption, the procedure described in Section 2 is carried out for all substrate edges. Additionally, trajectory parameters and the settling time of a rotation axis are added to the

assessment. The parameters for nanotube growth are not changed to be able to compare it to the results from Section 2.

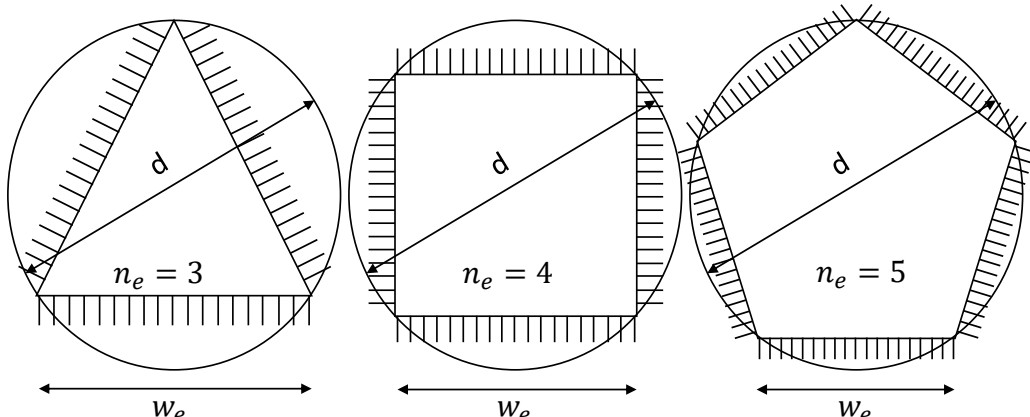

**Figure 9.** Different substrate geometries with polygonal shapes and the same circumcircle diameter $d$ characterized by the number of edges $n_e$. The width of each edge $w_e$ and the circumcircle diameter $d$ determine their size.

If Equations (8) and (9) are combined, the relationship between the number of edges $n_e$ and the number of cantilevers $n_C$ can be written as

$$d \cdot \sin \frac{\pi}{n_e} = (n_C - 1) \cdot p_C + w_C \tag{10}$$

Equation (10) points out the inverse relationship between the number of edges $n_e$ and the number of cantilevers per edge $n_C$. At one point while increasing the number of substrate edges $n_e$ the number of trenches, which is the number of cantilevers $n_C - 1$, will become zero and nanotubes could only be suspended across substrate edges. However, for practical reasons, like maintaining parallelism between cantilevers, this case is avoided in consequent investigations. With the condition that the number of cantilevers per substrate edge $n_C > 1$, the upper limit of the number of substrate edges $n_C$ can be calculated for a certain geometry.

According to Equation (10), the number of cantilevers per edge $n_C$ scale linearly with the circumcircle diameter $d$. Hence, the number of harvestable trenches $n_{harv}$ also scales with this diameter, since the nanotube distribution is uniform. This allows one to approximate the average number of harvestable trenches $n_{harv}$ for one edge and to assume that each substrate edge has a similar number. This effectively eliminates the time-consuming simulation of growth and selection for each individual edge.

Figure 10 exemplarily shows the influence of the different number of substrate edges $n_e$ and the trajectory parameters of the rotation axis onto productivity. The substrate design in terms of number of edges $n_e$ can be assessed by its required median exchange frequency $\bar{f}_e$. This value is the quotient of the median transfers per hour $\overline{P}$ and the median number of trenches $\bar{n}_{harv}$.

$$\bar{f}_e = \frac{\overline{P}}{\bar{n}_{harv}} \tag{11}$$

In Table 1, the median exchange frequency $\bar{f}_e$ indicates how many times the substrate must be exchanged per hour for continuous production. Taking the average required exchange time $\bar{t}_{ex}$ per whole growth substrate into account, the median effective productivity $\overline{P}_e$ can be calculated as

$$\overline{P}_e = \overline{P} \cdot (1 - \bar{f}_e \cdot \bar{t}_{ex}) \tag{12}$$

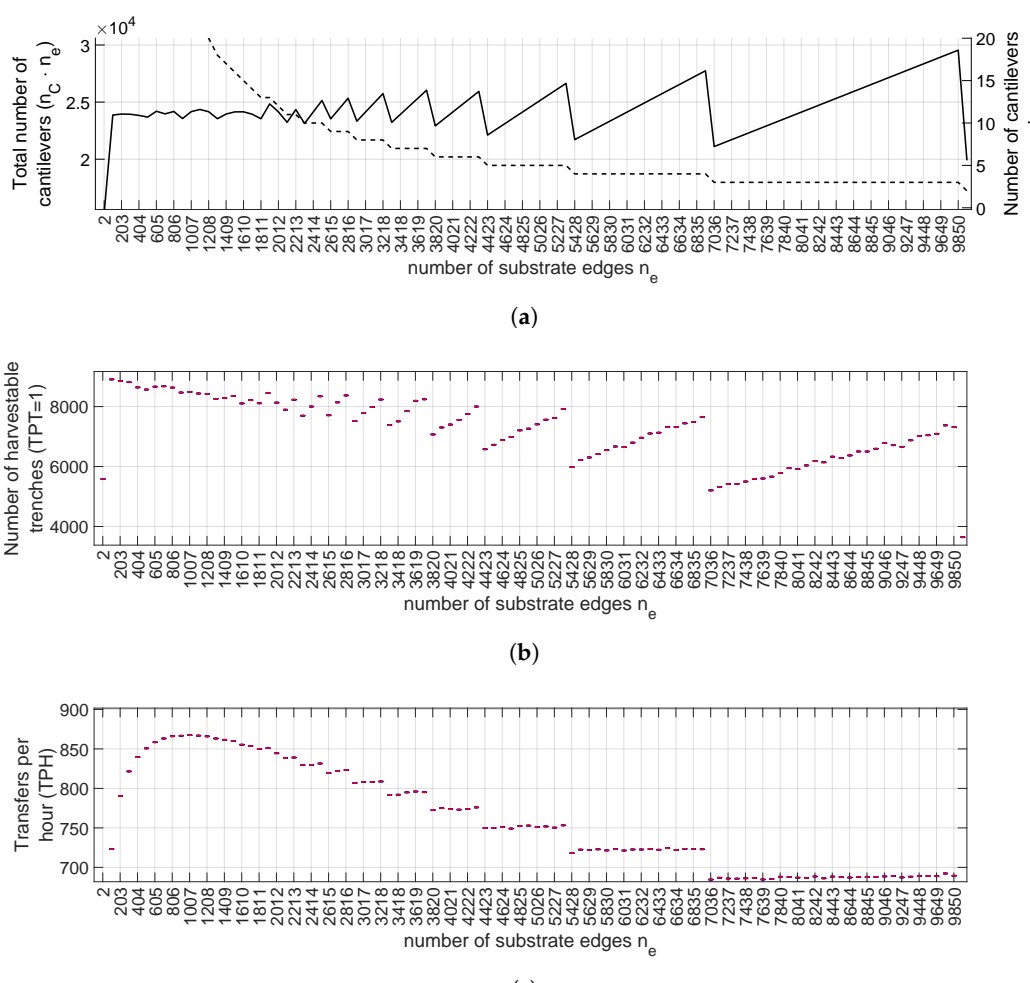

**Figure 10.** Simulation of varying polygonal substrate edges $n_e$ for a growth substrate with constant cantilever geometry $p_C = 5$ μm and $l_C = 60$ μm and a circumcircle diameter $d = 38$ mm. (**a**) The total number of cantilevers on a substrate $n_C \cdot n_e$ as solid line with its vertical axis on the left side and the number of cantilevers per substrate edge $n_C$ as a function of the number of substrate edges as dashed line and with its vertical axis on the right side. (**b**) Depending on the number of substrate edges, the count of harvestable trenches differs. This is significantly impacted by growth parameters and the number of cantilevers as shown in Figure 10a. (**c**) The frequency of substrate rotation changes with the number of substrate edges. This affects the assembly speed and is dependent on trajectory parameters and settling time of the rotation axis.

Taking an average required exchange time $\bar{t}_{ex}$ of 2 min into account, the preferable number of substrate edges $n_e$ would be 1007 according to the values from Table 1. This example is, however, only valid for the rotation axis with its chosen trajectory parameters and settling time. As depicted in Figure 11, a change in these parameters could cause a shift of the optimum substrate geometry with another number of edges $n_e$. A prototype of an assembly machine verifies the movement parameters used for the simulation and leads to a comparable effective productivity if overhead time is excluded. The proposed growth substrate design for automation already considers positioning-related factors, and a proof of concept has successfully been demonstrated. However, for future industrial utilization it is necessary to also consider the Mean Time Between Failure (MTBF) of various components to assess effective productivity.

**Table 1.** The required exchange frequencies and the median effective productivity $\overline{P_e}$ in TPH of substrates with different numbers of edges $n_e$ after Equation (11) and some values from Figure 10 with an average exchange duration $t_{ex}$ = 120 s.

| $n_e$ | 2 | 1007 | 2012 | 3017 | 4021 | 5026 | 6031 | 7036 | 8041 | 9046 |
|---|---|---|---|---|---|---|---|---|---|---|
| $\overline{P}$ | 181 | 867 | 844 | 808 | 774 | 751 | 721 | 685 | 687 | 689 |
| $\bar{n}_{harv}$ | 5584 | 8488 | 8132 | 7788 | 7400 | 7407 | 6646 | 5204 | 5912 | 6784 |
| $\bar{f}_e$ | 0.03 | 0.1 | 0.1 | 0.1 | 0.1 | 0.1 | 0.11 | 0.13 | 0.12 | 0.1 |
| $\overline{P_e}$ | 181 | 864 | 842 | 805 | 771 | 749 | 719 | 682 | 684 | 686 |

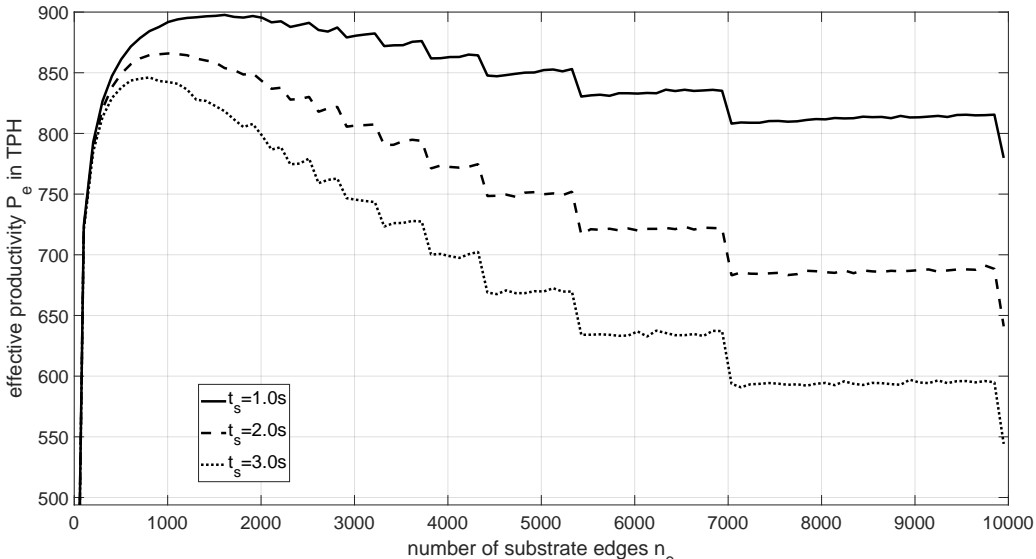

**Figure 11.** The effective productivity over the number of substrate edges for different settling times $t_s$ of the rotation axis.

## 4. Conclusions

The productivity of the mechanical dry transfer of carbon nanotubes from a growth substrate to a device substrate relies on the cantilever geometry. While the length of the cantilevers must be adjusted to the growth of the nanotubes and the desired number of tubes per trench, the cantilever pitch must be reduced to achieve the maximum productivity.

However, as positioning errors are present at all times, the cantilever pitch is limited on the lower end. With the total errors $\Delta X, \Delta Y, \Delta \varepsilon$ decreasing, higher device-trench-overlaps at higher angles can be achieved. This leads to an increase in the number of harvestable tubes and thus to a less frequent exchange of substrates. Furthermore, it is shown how the overlap factor $\eta_O$ affects the median transfer speed for various angles and cantilever pitches and lengths.

By adding a rotation axis, it becomes feasible to use substrates with multiple edges. This approach of increasing the total number of harvestable trenches leads to less frequent substrate exchanges. The analysis of the number of these edges shows that, depending on the trajectory parameters and the settling time of the rotation axis, the optimum number of substrate edges can be selected in order to reduce the substrate exchange frequency. Consequently, productivity is significantly amplified and also cost-effectiveness of the assembly process is improved. However, with each additional axis, system complexity increases. Therefore, the machine design with its configuration of axes must be considered already at early design stages when the required assembly motions are known.

**Author Contributions:** Conceptualization, A.B.; methodology, A.B.; software, A.B.; validation, A.B., S.W. and K.W.; formal analysis, A.B.; investigation, A.B. and S.W.; resources, A.B. and K.W.; data curation, A.B.; writing—original draft preparation, A.B.; writing—review and editing, A.B., S.W. and K.W.; visualization, A.B.; supervision, S.W. and K.W.; project administration, S.W. and K.W.; funding acquisition, K.W. All authors have read and agreed to the published version of the manuscript.

**Funding:** This research received no external funding.

**Data Availability Statement:** Data can be made available on request.

**Conflicts of Interest:** Authors Sascha Weikert and Konrad Wegener were employed by the company inspire AG. The remaining authors declare that the research was conducted in the absence of any commercial or financial relationships that could be construed as a potential conflict of interest.

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
