# Peer review of "Growth Substrate Geometry Optimization for the Productive Mechanical Dry Transfer of Carbon Nanotubes"

_processes, doi:10.3390/pr12050928_

Round 1
Reviewer 1 Report
Comments and Suggestions for Authors
First of all, I would like to express my sincere thanks for selecting our journal for your submission. The article is very well-written and meticulously prepared. Should you address the questions I have posed, I would be keen to review the article again.
- How was the simulation model validated against experimental data to ensure its accuracy in predicting carbon nanotube growth and transfer rates?
- How would variations in growth density across the substrate, due to experimental imperfections, affect the simulation outcomes?
- How do manipulator positioning errors and inaccuracies in substrate fabrication impact the reliability and efficiency of the carbon nanotube transfer process?
- Can the authors discuss the potential for automating the carbon nanotube transfer process and its impact on scalability and industrial applicability?
- What are the planned next steps in this research, particularly regarding more complex substrate geometries or dynamic environmental conditions?
- How critical are the clearance and overlap parameters in the assembly process, and what are the implications of underestimating these parameters on productivity and efficiency?
- Regarding optimisation and selection criteria, please briefly mention MCDM methods. These two studies will add depth to your article.
Polymeric Materials Selection for Flexible Pulsating Heat Pipe Manufacturing Using a Comparative Hybrid MCDM Approach
https://doi.org/10.3390/polym15132933
Application of MCDM method in material selection for optimal design: A review
https://doi.org/10.1016/j.rinma.2020.100115
- How does the addition of a rotation axis for substrates with multiple edges impact the overall system complexity and cost-effectiveness of the nanotube transfer process?
- How do the trajectory parameters and settling time of the rotation axis influence the selection of the optimum number of substrate edges for enhanced productivity?
- What are the practical limitations and challenges in implementing the proposed design for automation in a real-world manufacturing environment?
- Are there any trade-offs between increasing the total number of harvestable trenches and the complexity or cost of the substrate design, particularly with respect to multiple edges and rotation axes?
Author Response
Dear Reviewer #1,
Thank you for your thorough and prompt feedback. I am delighted to have discovered this special issue, as its topic aligns closely with the focus of my research.
In the attached file, I have addressed your comments and incorporated your suggestions, which are highlighted in yellow. I have endeavored to consider each aspect carefully and have revised the manuscript accordingly. In the resubmitted manuscript you will find that each change since the last version is also highlighted.
Please note that I also updated Figure 10. It not shows the results over much higher range of number of substrate edges. Those extended results also require a update of the values in Table 1.
Furthermore, I also added Figure 11, which shows the effects of the settling time of the rotation axis onto the effective productivity.
Sincere regards,
Andre Butzerin

Reviewer 2 Report
Comments and Suggestions for Authors
The geometry of the substrate could significantly affect the productivity of carbon nanotubes. The authors used a meta-modeling method to simulate the growth and selection of nanotubes on cantilevers. They investigated how the geometry of the cantilever affects the number of harvestable tranches and the transfer rate per hour. Higher device-trench-overlaps at higher angles showed effects on both the number of harvestable tubes and the transfer speed. The authors also studied the effect of the number of substrate edges on overall productivity. This systematic study provides an insight into the substrate geometry effect on the productivity of the suspended carbon nanotubes.
1. The authors did not provide a comparison with previous reports, either simulated or experimental. It would be valuable to compare the present method with the existing literature. In addition, are there any experiment evidences that support the present study (such as published one from other authors)?
2. Regarding equation (1), it would be helpful to include references or origins for clarity.
3. Geometry effect is discussed extensively in the main text, but there is a lack of comparison with previous studies. Including more discussion and comparison with previous research could enhance the results and provide a broader context for the findings.
Author Response
Dear Reviewer #2,
Thank you for your feedback!
In the attached file, I have addressed your comments and incorporated your suggestions, which are highlighted in yellow. In the resubmitted manuscript you will find that each change since the last version is also highlighted.
Please note that I also updated Figure 10. It not shows the results over much higher range of number of substrate edges. Those extended results also require a update of the values in Table 1.
Furthermore, I also added Figure 11, which shows the effects of the settling time of the rotation axis onto the effective productivity.
Sincere regards,
Andre Butzerin

Round 2
Reviewer 2 Report
Comments and Suggestions for Authors
The authors have addressed all the comments in the first version of the manuscript. They also added new simulated results and updated Figure 10 and Table 1. The effect of the setting time of the rotation axis has also been included in Figure 11. These changes have enriched the manuscript and enhanced the validity of science in the manuscript. Based on this, I believe that the manuscript can be accepted.